

# Impacts of updated reaction kinetics on the global GEOS-Chem simulation of atmospheric chemistry

Kelvin H. Bates[1,2], Mathew J. Evans[3,4], Barron H. Henderson[5], and Daniel J. Jacob[1]

[1]School of Engineering and Applied Sciences, Harvard University, Cambridge, MA 02138, USA
[2]NOAA Chemical Sciences Laboratory, Earth System Research Laboratories, & Cooperative Institute for Research in Environmental Sciences, University of Colorado, Boulder, CO 80305, USA
[3]Wolfson Atmospheric Chemistry Laboratories, Department of Chemistry, University of York, York, UK
[4]National Centre for Atmospheric Science, University of York, York, UK
[5]U.S. Environmental Protection Agency, Research Triangle Park, NC 27711, USA

**Correspondence:** Kelvin Bates (kelvin.bates@noaa.gov)

**Abstract.**

We updated the chemical mechanism of the GEOS-Chem global 3-D model of atmospheric chemistry to include new recommendations from the JPL chemical kinetics Data Evaluation 19-5 and from IUPAC, and to balance carbon and nitrogen. We examined the impact of these updates on the GEOS-Chem version 14.0.1 simulation. Notable changes include: eleven
updates to reactions of reactive nitrogen species, resulting in a 7% net increase in the stratospheric $NO_x$ (NO + $NO_2$) burden; an updated CO + OH rate formula leading to a 2.7% reduction in total tropospheric CO; adjustments to the rate coefficient and branching ratios of propane + OH, leading to reduced tropospheric propane (-17%) and increased acetone (+3.5%) burdens; a 41% increase in the tropospheric burden of peroxyacetic acid due to a decrease in the rate coefficient for its reaction with OH, further contributing to reductions in peroxyacetyl nitrate (PAN; -3.8%) and acetic acid (-3.4%); and a number of minor
adjustments to halogen radical cycling. Changes to the global tropospheric burdens of other species include -0.7% for ozone, +0.3% for OH (-0.4% for methane lifetime against oxidation by tropospheric OH), +0.8% for formaldehyde, and -1.7% for $NO_x$. The updated mechanism reflects the current state of the science including complex chemical dependencies of key atmospheric species on temperature, pressure, and concentrations of other compounds. The improved conservation of carbon and nitrogen will facilitate future studies of their overall atmospheric budgets.

## 1   Introduction

Chemical reactions underpin the trace gas composition of the atmosphere, removing some pollutants and climate forcing species while forming others. Accurate representation of reaction rate coefficients and product distributions of these reactions is crucial for atmospheric chemistry models. The NASA Jet Propulsion Laboratory (JPL) regularly assembles a panel of experts to assess and compile new data for rate constants and other key reaction parameters relevant to atmospheric chemistry. This
panel produces data evaluations entitled "Chemical Kinetics and Photochemical Data for Use in Atmospheric Studies". Recommendations are predominantly based on laboratory measurements, although they may also be informed by computational





studies or by analogy to similar reactions when required, and are not adjusted to fit ambient observations. Where assessed experimental studies disagree on reaction parameters, the panel attempts to reconcile and/or arbitrate differences and provide estimated uncertainty bounds. A parallel evaluation process is conducted by the International Union of Pure and Applied
Chemistry (IUPAC) with more emphasis on organic chemistry (Atkinson et al., 2006).

Here, we update the chemical mechanism in the GEOS-Chem global atmospheric chemistry model with recommendations from the latest JPL Data Evaluation (Burkholder et al., 2020) and recent IUPAC updates (Mellouki et al., 2021). The GEOS-Chem model is used by hundreds of research groups worldwide for global and regional studies of tropospheric and stratospheric chemistry (http://geos-chem.org), and is also used as a chemistry module in meteorological and climate models (Hu et al.,
2018; **?**; Fritz et al., 2022). We enforce carbon and nitrogen conservation in a further 63 reactions to facilitate budget analyses (Safieddine et al., 2017). We examine the effects of these different changes on the GEOS-Chem simulation as documentation for their inclusion in the standard version of the model.

## 2 GEOS-Chem chemical mechanism updates

The GEOS-Chem model includes a detailed mechanism for oxidant-aerosol chemistry in the troposphere and stratosphere.
Kinetics and products in the standard GEOS-Chem chemical mechanism generally follow JPL recommendations, supplemented by those of IUPAC (Atkinson et al., 2006). The GEOS-Chem mechanism in version 14.0.1 (DOI 10.5281/zenodo.7271974), which we use as reference here, is mostly based on older JPL and IUPAC recommendations, though previous updates have been piecemeal. The mechanism goes beyond these recommendations for the oxidation of isoprene (Marais et al., 2016; Bates and Jacob, 2019), monoterpenes (Fisher et al., 2016), aromatics (Bates et al., 2021), and for tropospheric halogen chemistry
Wang et al. (2021). Photolysis frequencies are calculated with the Fast-JX code (Bian and Prather, 2002) as implemented in GEOS-Chem by Mao et al. (2010) for the troposphere and Eastham et al. (2014) for the stratosphere.

Here we focus on gas-phase thermal chemistry updates as given by JPL and IUPAC. We do not include photolysis updates, as these are not significant, or aerosol and cloud chemistry, for which current GEOS-Chem treatments are documented in Holmes et al. (2019); Shah et al. (2020); Wang et al. (2021). A total of 63 gas-phase thermal reactions (out of the 650 in GEOS-Chem
version 14.0.1, in addition to 157 photolyses and 107 aerosol and cloud reactions) are updated with new rate coefficients and/or product distributions, including one newly added reaction. Table 1 lists the reactions updated with the old and the new rate coefficients and products given, and Figures S1-S9 in the supplementary material show the impact of these updates on rate coefficients as a function of temperature and/or pressure. The following paragraphs provide a summary of the changed rate coefficients and product yields, organized by chemical family as in Table 1. All updates are from the JPL Data Evaluation
unless otherwise noted.

Eleven updated reaction rate coefficients correspond to inorganic reactions in the reactive nitrogen ($NO_z$) family, with new values derived predominantly from changes to the methods of combining data from previous studies. Most notable among these updates is a reversal in the temperature dependence from positive to negative for the reaction between nitrous acid (HONO) and OH, resulting in a 40% increase in the rate coefficient at 300 K rising to a 300% increase at 200 K. The new




assessment prioritizes the temperature dependence measured by Burkholder et al. (1992) over that of Jenkin and Cox (1987), which had previously been used. Rate coefficient formulas for the two pressure- and temperature-dependent branches of of the O + NO$_2$ reaction – association to yield NO$_3$ and activation to yield NO + O$_2$ – were rebalanced, resulting an increases to the former branch of 19-21% and a decrease to the latter of 10-18% (producing an overall decrease of 6-9) under typical conditions between altitudes of 0 km (288 K, 1013 hPa) and 10 km (240 K, 270 hPa). The combined termolecular activation

and association reactions of OH + HNO$_3$, both yielding NO$_3$ + H$_2$O, were also reanalyzed based on new experimental work by Winiberg et al. (2018) and Dulitz et al. (2018), resulting in changes to its rate coefficient of -1% to +4% over the range of atmospheric conditions. Other updates in this family include a rate coefficient decrease for OH + NO$_3$ (-9%) and increase for NO$_3$ + O (+30%); weaker temperature dependencies and adjusted rate constants for HO$_2$ + NO, NO + NO$_3$, and N + O$_2$, resulting in changes to their coefficients over the range 200–300 K of -1% – +1%, -10% – -2%, and +109% – -1% respectively;

and stronger temperature dependencies for NO$_2$ + NO$_3$ (-33% – -25%) and OH + HNO$_4$ (+9% – -25%).

Thirteen of the changes are updates to branching ratios in reactions of O$^1$D with HBr, HCl, and halogenated organic compounds. These adjustments are generally minor, shifting the branching between the O + XR and OX + R product channels (where X stands in for Cl or Br and R stands in for H or an organic group) by a few percent. We also add the newly included O$^1$D + CH$_3$Cl reaction and update the temperature dependencies of the O$^1$D + CFC114 and O$^1$D + CFC115 reactions to

the JPL recommendations based on Baasandorj et al. (2013), resulting in increases of between 18% and 35% to their rate coefficients over the range 200–300 K.

Four updates apply to the rate coefficients of organic ozonolysis reactions. While three of these (O$_3$ + ethene, propene, and methacrolein) are only small adjustments to temperature dependence, resulting in rate coefficient increases of 2–12% over the range 200–300 K, the isoprene ozonolysis reaction in GEOS-Chem did not previously have a temperature dependence. We

include that now per JPL recommendation, which considerably slows ozonolysis at low temperatures (-96% at 200 K, -44% at 273 K) and accelerates it when T > 297 K.

Ten rate coefficients of OH + organic reactions are changed – five based on JPL recommendations and five from IUPAC. From JPL, the notable update is to the two C$_3$H$_8$ + OH branches (primary and secondary abstraction). The two branches have different temperature dependencies, a feature which had previously been included in GEOS-Chem, but the reanalysis in this

JPL Data Evaluation changes both dependencies considerably, reversing the dependence of the secondary abstraction. Overall, the importance of secondary abstraction branch increases; its rate coefficient goes up 300% at 200 K but barely changes at 300, while the primary goes up 140% at 200 but drops 30% at 300. (This is after accounting for the apparent typo in the primary abstraction rate constant from the JPL Data Evaluation, which would increase its rate coefficient by a factor of 10). The other JPL-derived revisions include a reduction in the OH oxidation rate coefficient for C$_{3+}$ alcohols ("ROH") by 4% and

weaker temperature dependencies for OH + CH$_2$Cl$_2$ and CH$_3$Cl, changing their rate coefficients by +2% – -9% and +37% – -3% respectively over the temperature range 200–300 K. The IUPAC-derived updates include the addition of a temperature dependence for the methylglyoxal + OH reaction based on Baeza-Romero et al. (2007) and Tyndall et al. (1995), changing its rate coefficient by +125% at 200 K and -14% at 300 K; minor changes to the temperature dependencies of the methyl ethyl ketone + OH rate coefficient (-17% – -7% at 200–300 K) and to both branches of the hydroxyacetone + OH reaction (0% –



-2%), and a factor of 30-50 downward revision to the OH + peroxyacetic acid (PAA) rate coefficient based on the experimental and theoretical results of Berasategui et al. (2020).

Five rate coefficients of reactions between the Cl radical and organic compounds are updated. The Cl + acetone rate coefficient now has a smaller temperature dependence and faster rate at temperatures below 251 K, resulting in a slower reaction by 22% at 300 K but a faster reaction by 49% at 200 K than previously. The Cl + $CH_2O$ reaction has the same temperature

dependence, but its rate constant is revised upward by 11%. The other three changes are to Cl + chloromethane compounds; the Cl + $CH_2Cl_2$ rate constant, though unchanged in Publication 19-5 from previous JPL data evaluations, may have been erroneously low in GEOS-Chem, because we find it requires increasing by a factor of 10.2–13.3 over the range 200–300 K to match the recommendation. The updates to the Cl + $CH_3Cl$ and Cl + $CHCl_3$ coefficients are much smaller, changing by +3% – 0% and -3% – -6% respectively over the range 200–300 K.

Five revisions apply to reactions of organic peroxy ($RO_2$) radicals, including increases in the rate coefficients of the self reaction of the peroxyacetyl radical ($CH_3CO_3$) by 14%, the reaction of NO with the acetone-derived peroxy radical $CH_3C(O)CH_2O_2$ by 4%, the reaction of the ethylperoxy radical ($CH_3CH_2O_2$) with $HO_2$ (+1%), and other functionalized $C_2$ peroxy radicals + $HO_2$ (+1%). These updates also include a slight increase in the temperature- and pressure-dependent peroxyacetyl nitrate (PAN) formation rate (the termolecular reaction of $CH_3CO_3 + NO_2$) by about 2% under typical conditions

between altitudes of 0 km (288 K, 1013 hPa) and 10 km (240 K, 270 hPa). However, the PAN equilibrium remains unchanged in this JPL assessment, so we also adjust the dissociation reaction rate coefficient accordingly.

Four updates pertain to the fates of Criegee Intermediates. Some of these reactions are newly included in the JPL Data Evaluation, and had previously received scant attention in GEOS-Chem; they were generally added selectively to the mechanism either to provide a source of HCOOH (Millet et al., 2015) or to complete the isoprene oxidation cascade (Bates and

Jacob, 2019). Updates include substantial increases in the rate coefficients of $CH_2OO + NO_2$ (+4250%) and $CH_3CHOO + SO_2$ (+377%), a slight increase to $CH_2OO + SO_2$ (+3%), and a sharp decrease to the rate coefficient of $CH_2OO + H_2O$ (-84%).

Six changes are made to reactions in the iodine radical chemistry scheme, most notably a downward revision by 54% to the IO + ClO rate constant. The temperature dependence of the IO + BrO reaction is increased, resulting in changes to its rate coefficient of +28% at 200 K ranging to -16% at 300 K. Conversely, temperature dependencies are weakened for the IO +

NO and I + $O_3$ reactions, resulting in changes of +6% – -1% and -10% – -9% respectively over the range 200-300 K. Slight adjustments to the I + NO and I + $NO_2$ rate coefficients result in decreases of <1% and <3% respectively.

Five other miscellaneous rate coefficient changes complete the list. First, a revision to the formula for the termolecular reaction of $SO_2$ with OH results in pressure- and temperature-dependent rate coefficient decreases of 0–5% under typical conditions between altitudes of 0 km (288 K, 1013 hPa) and 10 km (240 K, 270 hPa). Similar revisions to the formula for

the termolecular H + $O_2$ reaction and the combined activation and association reactions of OH + CO result in rate coefficient increases under the same range of atmospheric conditions of 23–34% and 3-5% respectively. Finally, a weaker temperature dependence for the reaction of OCS with OH renders the rate coefficient nearly unchanged at 300 K but 10% faster at 250 K and 25% faster at 200 K.



The product distributions of a further 63 reactions listed in Table S2 were adjusted so as to balance carbon and nitrogen
between the reactants and products. Previously, $CO_2$ had not explicitly been represented as a product in many reactions,
while others were imbalanced due to rounding errors or uncertainties in product branching ratios. The updated reactions were
balanced predominantly by adding $CO_2$ as a coproduct, adjusting branching ratios slightly to offset rounding errors, or adding
lumped organic products (e.g. "RCHO" for $C_{3+}$ aldehydes, which carries 3 carbon atoms) to account for products whose
specific structure is unknown. There remain 54 reactions (Table S3) that are imperfectly carbon- or nitrogen-balanced; most
are (a) in the monoterpene oxidation submechanism (Fisher et al., 2016); (b) reactions of (H)CFCs, which play a minor role in
the atmospheric carbon budget; and (c) reactions coupling the gas-phase organic mechanism to the formation of non-specific
particle-phase species (e.g., a single aerosol-phase isoprene-derived nitrate species carries five carbons and one nitrogen, but is
formed via uptake of species with 4-5 carbons and 1-2 nitrogens). Balancing these reactions will be the focus of future efforts.

Table 1: GEOS-Chem reactions updated per JPL and IUPAC recommendations[a]

| Reactants | Products (old mechanism) | Rate coefficient[b] (old mechanism) | New rate coefficient[b] and/or products |
|---|---|---|---|
| *$NO_z$ reactions* | | | |
| $OH + NO_3$ | $HO_2 + NO_2$ | $2.2 \times 10^{-11}$ | $2.0 \times 10^{-11}$ |
| $NO_2 + NO_3$ | $NO + NO_2 + O_2$ | $4.5 \times 10^{-14} \times e^{-1260/T}$ | $4.35 \times 10^{-14} \times e^{-1335/T}$ |
| $O + NO_3$ | $NO_2 + O_2$ | $1.0 \times 10^{-11}$ | $1.3 \times 10^{-11}$ |
| $OH + HNO_2$ | $NO_2 + H_2O$ | $1.8 \times 10^{-11} \times e^{-390/T}$ | $3.0 \times 10^{-12} \times e^{250/T}$ |
| $OH + HNO_4$ | $NO_2 + O_2 + H_2O$ | $1.3 \times 10^{-12} \times e^{380/T}$ | $4.5 \times 10^{-13} \times e^{610/T}$ |
| $HO_2 + NO$ | $OH + NO_2$ | $3.3 \times 10^{-12} \times e^{270/T}$ | $3.44 \times 10^{-12} \times e^{260/T}$ |
| $N + O_2$ | $NO + O$ | $1.5 \times 10^{-11} \times e^{-3600/T}$ | $3.3 \times 10^{-12} \times e^{-3150/T}$ |
| $NO + NO_3$ | $2\,NO_2$ | $1.5 \times 10^{-11} \times e^{170/T}$ | $1.7 \times 10^{11} \times e^{125/T}$ |
| $OH + HNO_3\ [+ M]$ | $NO_3 + H_2O\ [+ M]$ | $f_x(2.41\text{e-}14, 2.69\text{e-}17, 6.51\text{e-}34)^c$ | $f_a(3.9\text{e-}31, 7.2, 1.5\text{e-}13, 4.8, 3.7\text{e-}14, -240)^c$ |
| $O + NO_2$ | $NO + O_2$ | $5.1 \times 10^{-12} \times e^{210/T}$ | $f_b(3.4\text{e-}31, 1.6, 2.3\text{e-}11, 0.2, 5.3\text{e-}12, -200)^c$ |
| $O + NO_2\ [+ M]$ | $NO_3\ [+ M]$ | $f_t(2.5\text{e-}31, 1.8, 2.2\text{e-}11, 0.7)^c$ | $f_t(3.4\text{e-}31, 1.6, 2.3\text{e-}11, 0.2)^c$ |
| *$O^1D$-halogen reactions* | | | |
| $O^1D + HCl$ | $0.67(Cl + OH) + 0.24(ClO + H) + 0.09(HCl + O)$ | $1.5 \times 10^{-10}$ | $0.66(Cl + OH) + 0.22(ClO + H) + 0.12(HCl + O)$ |
| $O^1D + HBr$ | $0.65(Br + OH) + 0.2(HBr + O) + 0.15(BrO + H)$ | $1.5 \times 10^{-10}$ | $0.6(Br + OH) + 0.2(HBr + O) + 0.2(BrO + H)$ |
| $O^1D + CHBr_3$ | $1.36\,Br + 0.68\,BrO + 0.32(CHBr_3 + O)$ | $6.6 \times 10^{-10}$ | $1.4\,Br + 0.7\,BrO + 0.3(CHBr_3 + O)$ |
| $O^1D + CCl_4$ | $2.58\,Cl + 0.86\,ClO + 0.14(CCl_4 + O)$ | $3.3 \times 10^{-10}$ | $2.37\,Cl + 0.79\,ClO + 0.21(CCl_4 + O)$ |
| $O^1D + CFC11$ | $1.76\,Cl + 0.88\,ClO + 0.12(CFC11 + O)$ | $2.3 \times 10^{-10}$ | $1.8\,Cl + 0.9\,ClO + 0.1(CFC11 + O)$ |
| $O^1D + CFC113$ | $1.5\,Cl + 0.75\,ClO + 0.25(CFC113 + O)$ | $2.32 \times 10^{-10}$ | $1.8\,Cl + 0.9\,ClO + 0.1(CFC113 + O)$ |
| $O^1D + CFC114$ | $0.75(Cl + ClO) + 0.25(CFC114 + O)$ | $1.3 \times 10^{-10} \times e^{-25/T}$ | $1.3 \times 10^{-10} \times e^{25/T}$; $0.95\,Cl + 0.85\,ClO + 0.1(CFC114 + O)$ |
| $O^1D + CFC115$ | $0.7(CFC115 + O) + 0.3\,ClO$ | $5.4 \times 10^{-11} \times e^{-30/T}$ | $5.4 \times 10^{-11} \times e^{30/T}$; $0.86\,ClO + 0.14(CFC115 + O)$ |

*Continued on next page*





*Continued from previous page*

| Reactants | Products (old mechanism) | Rate coefficient[b] (old mechanism) | New rate coefficient[b] and/or products |
|---|---|---|---|
| $O^1D$ + HCFC22 | 0.55 ClO + 0.28(HCFC22 + O) + 0.17 Cl | $1.02 \times 10^{-10}$ | 0.56 ClO + 0.25(HCFC22 + O) + 0.19 Cl + 0.05 OH |
| $O^1D$ + HCFC142b | 0.74 ClO + 0.26(HCFC142b + O) | $2.0 \times 10^{-10}$ | 0.65 ClO + 0.35(HCFC142b + O) |
| $O^1D$ + H1211 | 0.36(H1211 + O) + 0.33(ClO + Br) + 0.31(Cl + BrO) | $1.5 \times 10^{-10}$ | 0.35(H1211 + O) + 0.34(ClO + Br) + 0.31(Cl + BrO) |
| $O^1D$ + H1301 | 0.59(H1301 + O) + 0.41 BrO | $1.0 \times 10^{-10}$ | 0.55(H1301 + O) + 0.45 BrO |
| $O^1D$ + $CH_3Cl$ | N/A | N/A | $2.6 \times 10^{-10}$; 0.9 $CH_3OO$ + 0.46 ClO + 0.35 Cl + 0.1(O + $CH_3Cl$) + 0.09 H |
| *Ozonolysis reactions* | | | |
| Ethene + $O_3$ | $CH_2O$ + $CH_2OO$ | $9.1 \times 10^{-15} \times e^{-2580/T}$ | $1.2 \times 10^{-14} \times e^{-2630/T}$ |
| Propene + $O_3$ | 0.5 $CH_3CHO$ + 0.5 $CH_2O$ + ... | $5.5 \times 10^{-15} \times e^{-1880/T}$ | $6.5 \times 10^{-15} \times e^{-1900/T}$ |
| Methacrolein + $O_3$ | 0.88 $CH_3C(O)CHO$ + ... | $1.4 \times 10^{-15} \times e^{-2100/T}$ | $1.5 \times 10^{-15} \times e^{-2110/T}$ |
| Isoprene + $O_3$ | 0.827 $CH_2O$ + 0.58 $CH_2OO$ + ... | $1.3 \times 10^{-17}$ | $1.1 \times 10^{-14} \times e^{-2000/T}$ |
| *OH + organic reactions* | | | |
| Propane + OH | $CH_3CH(OO)CH_3$ | $f_P(5.87,0.64,-816)^c$ | $8.54 \times 10^{-13} \times e^{-19/T} \times (\frac{298}{T})^{1.54}$ |
| Propane + OH | $CH_3CH_2CH_2OO$ | $f_P(0.17,-0.64,816)^c$ | $1.97 \times 10^{-12} \times e^{-675/T} \times (\frac{298}{T})^{1.23}$ |
| $C_{3+}$ alcohol + OH | $C_{3+}$ aldehyde + $HO_2$ | $4.6 \times 10^{-12} \times e^{70/T}$ | $4.4 \times 10^{-12} \times e^{70/T}$ |
| $CH_2Cl_2$ + OH | 2 Cl + $HO_2$ | $2.61 \times 10^{-12} \times e^{-944/T}$ | $1.92 \times 10^{-12} \times e^{-880/T}$ |
| $CHCl_3$ + OH | 3 Cl + $HO_2$ | $4.69 \times 10^{-12} \times e^{-1134/T}$ | $2.2 \times 10^{-12} \times e^{-920/T}$ |
| $CH_3C(O)CHO$ + OH | $CH_3CO_3$ + CO | $1.5 \times 10^{-11}$ | [d] $1.9 \times 10^{-12} \times e^{575/T}$ |
| $CH_3C(O)CH_2CH_3$ + OH | $CH_3C(O)CH_2CH_2OO$ | $1.3 \times 10^{-12} \times e^{-25/T}$ | [d] $1.5 \times 10^{-12} \times e^{-90/T}$ |
| $CH_3CO_3H$ + OH | $CH_3CO_3$ | $6.13 \times 10^{-13} \times e^{200/T}$ | [d] $3.0 \times 10^{-14}$; 0.78 $CH_3CO_3$ + 0.22(OH + $CO_2$ + $CH_2O$) |
| $CH_3C(O)CH_2OH$ + OH | $CH_3C(O)CHO$ + $HO_2$ | $f_{HA}(2.15e\text{-}12,305)^c$ | [d] $f_{HA}(2.0e\text{-}12,320)^c$ |
| $CH_3C(O)CH_2OH$ + OH | 0.5($CH_3CO_2H$ + HCOOH) + ... | $f_{HB}(2.15e\text{-}12,305)^c$ | [d] $f_{HB}(2.0e\text{-}12,320)^c$ |
| *Cl + organic reactions* | | | |
| Cl + $CH_2O$ | CO + HCl + $HO_2$ | $7.32 \times 10^{-11} \times e^{-30/T}$ | $8.1 \times 10^{-11} \times e^{-30/T}$ |
| Cl + Acetone | HCl + $CH_3C(O)CH_2OO$ | $7.7 \times 10^{-11} \times e^{-1000/T}$ | $1.63 \times 10^{-11} \times e^{-610/T}$ |
| Cl + $CH_3Cl$ | CO + 2 HCl + $HO_2$ | $2.17 \times 10^{-11} \times e^{-1130/T}$ | $2.03 \times 10^{-11} \times e^{-1110/T}$ |
| Cl + $CH_2Cl_2$ | CO + HCl + $HO_2$ + 2 Cl | $1.24 \times 10^{-12} \times e^{-1070/T}$ | $7.4 \times 10^{-12} \times e^{-910/T}$ |
| Cl + $CHCl_3$ | CO + HCl + $HO_2$ + 3 Cl | $3.77 \times 10^{-12} \times e^{-1011/T}$ | $3.3 \times 10^{-12} \times e^{-990/T}$ |
| *Peroxy radical ($RO_2$) reactions* | | | |
| $CH_3CO_3$ + $CH_3CO_3$ | 2 $CH_3OO$ | $2.5 \times 10^{-12} \times e^{500/T}$ | $2.9 \times 10^{-12} \times e^{500/T}$ |
| $CH_3C(O)CH_2OO$ + NO | $CH_3CO_3$ + $CH_2O$ + $NO_2$ | $2.8 \times 10^{-12} \times e^{300/T}$ | $2.9 \times 10^{-12} \times e^{300/T}$ |
| $CH_3CH_2OO$ + $HO_2$ | $CH_3CH_2OOH$ | $7.4 \times 10^{-13} \times e^{700/T}$ | $7.5 \times 10^{-13} \times e^{700/T}$ |
| $OTHRO2^e$ + $HO_2$ | $CH_3CH_2OOH$ | $7.4 \times 10^{-13} \times e^{700/T}$ | $7.5 \times 10^{-13} \times e^{700/T}$ |
| $CH_3CO_3$ + $NO_2$ [+ M] | PAN [+ M] | $f_t(9.7e\text{-}29,5.6,9.3e\text{-}12,1.5)^c$ | $f_t(7.3e\text{-}29,4.1,9.5e\text{-}12,1.6)^c$ |
| *Criegee Intermediate reactions* | | | |
| $CH_2OO$ + $NO_2$ | $CH_2O$ + $NO_3$ | $1.0 \times 10^{-15}$ | $4.25 \times 10^{-12}$ |
| $CH_2OO$ + $SO_2$ | $CH_2O$ + $SO_4$ | $3.7 \times 10^{-11}$ | $3.8 \times 10^{-11}$ |
| $CH_2OO$ + $H_2O$ | 0.73 HMHP + 0.21 HCOOH + ... | $1.7 \times 10^{-15}$ | $2.8 \times 10^{-16}$ |
| $CH_3CH_2OO$ + $SO_2$ | $CH_3CHO$ + $SO_4$ | $7.0 \times 10^{-14}$ | $2.65 \times 10^{-11}$ |
| *Iodine reactions* | | | |
| IO + NO | I + $NO_2$ | $9.1 \times 10^{-12} \times e^{240/T}$ | $8.6 \times 10^{-12} \times e^{230/T}$ |
| IO + ClO | 0.801 I + 0.56 OClO + ... | $8.93 \times 10^{-12} \times e^{280/T}$ | $4.82 \times 10^{-12} \times e^{280/T}$ |





*Continued from previous page*

| Reactants | Products (old mechanism) | Rate coefficient[b] (old mechanism) | New rate coefficient[b] and/or products |
|---|---|---|---|
| IO + BrO | Br + 0.8 OIO + ... | $1.5 \times 10^{-11} \times e^{510/T}$ | $5.5 \times 10^{-12} \times e^{760/T}$ |
| I + O$_3$ | IO + O$_2$ | $2.3 \times 10^{-11} \times e^{-870/T}$ | $2.0 \times 10^{-11} \times e^{-830/T}$ |
| I + NO [+ M] | INO [+ M] | $f_t$(1.8e-32,1,1.77e-11,0)[c] | $f_t$(1.8e-32,1,1.7e-11,0)[c] |
| I + NO$_2$ [+ M] | IONO [+ M] | $f_t$(3e-31,1,6.6e-11,0,0.63)[c] | $f_t$(3e-31,1,6.6e-11,0,0.6)[c] |
| *Other reactions* | | | |
| OCS + OH | CO$_2$ + SO$_2$ | $1.1 \times 10^{-13} \times e^{-1200/T}$ | $7.2 \times 10^{-14} \times e^{-1070/T}$ |
| CO + OH [+ M] | HO$_2$ + CO$_2$ [+ M] | $f_c$(5.9e-33,1.1e-12,1.5e-13,2.1e9)[c] | $f_a$(6.9e-33,2.1,1.1e-12,-1.3,1.85e-13,65)[c] |
| SO$_2$ + OH [+ M] | SO$_4$ + HO$_2$ [+ M] | $f_t$(3.3e-31,4.3,1.6e-12,0)[c] | $f_t$(2.9e-31,4.1,1.7e-12,-0.2)[c] |
| H + O$_2$ [+ M] | HO$_2$ [+ M] | $f_t$(4.4e-32,1.3,7.5e-11,-0.2)[c] | $f_t$(5.3e-32,1.8,9.5e-11,-0.4)[c] |
| PAN [+ M] | CH$_3$CO$_3$ + NO$_2$ [+ M] | $f_d$(9.7e-29,5.6,9.3e-12,1.5)[c] | $f_d$(7.3e-29,4.1,9.5e-12,1.6)[c] |

[a] Updates are from JPL Data Evaluation recommendations unless denoted otherwise by a table footnote in Column 4; GEOS-Chem names for chemical species listed here by formula or common name are given in Table S1; [b] In units cm$^3$ molecule$^{-1}$ s$^{-1}$ unless otherwise noted; $T$ is temperature in K. [c] Formulas for the rate coefficients of propane + OH ($f_P$), hydroxyacetone + OH ($f_{HA}$ & $f_{HB}$), termolecular reactions ($f_t$), dissociations ($f_d$, units s$^{-1}$), activation reactions ($f_a$ & $f_b$), and previous GEOS-Chem parameterizations of OH + HNO$_3$ ($f_x$) and OH + CO ($f_c$) are given in Section S1; [d] Per IUPAC recommendations; [e] OTHRO2 represents functionalized C$_2$ peroxy radicals aside from CH$_3$CH$_2$OO.

## 3 Model simulations

We use version 14.0.1 of GEOS-Chem Classic (http://www.geos-chem.org, DOI: 10.5281/zenodo.7271974) driven by NASA MERRA-2 assimilated meteorological data to simulate the impacts of the updated rate coefficients on trace gas concentrations and budgets. Detailed gas and aerosol chemistry is computed throughout the troposphere and stratosphere at 30 minute time intervals using a fourth-order Rosenbrock kinetic solver implemented with the Kinetic PreProcessor (KPP) version 3.0 (Lin et al., 2023). Emissions are calculated at 30 minute time steps using the standard GEOS-Chem emissions from the Harmonized Emissions Component (HEMCO) version 3.0 (Lin et al., 2021). This includes anthropogenic emissions from the Community Emissions Data System (CEDS) inventory (Hoesly et al., 2018), biomass burning emissions from the Global Fire Emissions Database (GFED) version 4 (van der Werf et al., 2010), and biogenic VOC emissions from the Model of Emissions of Gases and Aerosols from Nature (MEGAN) version 2.1 (Guenther et al., 2012) as implemented by Hu et al. (2015) and calculated offline to improve reproducibility across scales (Weng et al., 2020). Methane is treated as an advected and reactive species, but without emissions; instead, surface concentrations are prescribed based on measured monthly means (Murray, 2016).

To calculate the changes in burdens and budgets of atmospheric species described in the following section, we perform two simulations: one with the unchanged chemical mechanism from GEOS-Chem classic version 14.0.1 ("base") and another with the mechanism altered to include the updates listed in Tables 1, S2, and S4 ("updated"). Both simulations are performed at $2° \times 2.5°$ horizontal resolution over 72 vertical levels, are initialized from the same generic concentration fields, and are run for two years (1 January 2017 to 1 January 2019). The first year is to remove the effect of the common initialization. We use output from the second year to compute the changes described below.



## 4 Impacts on species concentrations

Figures 1-5 show the impacts of the rate coefficient and product yield updates on annually averaged surface mixing ratios,
zonally averaged mixing ratios, and total tropospheric and stratospheric burdens of selected species. These changes are also
described and explained in the following paragraphs. A more complete list of impacts on annually averaged tropospheric mass
burdens for all species in GEOS-Chem that change by >1% can be found in Table S5, and a similar list for stratospheric changes
can be found in Table S6.

Figure 1 shows the effects of the mechanism changes on ozone, $HO_x$ (= OH + $HO_2$), and $H_2O_2$. Total tropospheric ozone
decreases by 0.7% in the updated mechanism, corresponding to a surface decrease of 0.5 ppb in the extratropics, driven in part
by reduced PAN-driven $NO_x$ transport (described below) and in part by faster organic ozonolysis reactions. Over East Asia,
where faster CO + OH and $C_3H_8$ + OH reactions increase radical cycling, and tropical forests, where isoprene ozonolysis is
slowed and $NO_x$ is increased due to better nitrogen conservation in the reaction forming methyl vinyl ketone hydroxynitrate
(MVKN), surface ozone rises by 0.1-0.3 ppb. The upper troposphere and lower stratosphere experience much stronger ozone
decreases of up to 5% in the extratropics from higher $NO_x$-driven losses brought about by changes to the $HNOz$ + OH (z=2–4)
reactions, but this is partially balanced by an ozone increase at 50-100 hPa driven by the updated N + $O_2$ reaction, leading to a
net decrease of only 0.7% in stratospheric ozone.

In contrast to the decreased ozone, tropospheric $HO_x$ and $H_2O_2$ burdens all increase slightly (OH +0.3%, $HO_2$ +1.0%, $H_2O_2$
+1.3%). Increased tropospheric OH is driven largely by the reduced PAA + OH rate coefficient, particularly in the upper tropo-
sphere, which is partially offset by the changes to CO + OH and $HNO_z$ + OH rate coefficients. Higher OH leads to increases in
the methane loss rate, causing an overall 0.4% decrease in the methane lifetime against oxidation by tropospheric OH. Higher
tropospheric $HO_2$ and $H_2O_2$ are driven largely by the accelerated CO + OH rate, with stronger increases in the upper tropo-
sphere due to changes in $HNOz$ + OH and over regions with strong biogenic influence due to the revised nitrogen conservation
from MVKN. Higher $HO_2$ over forests increases the proportion of isoprene-derived peroxy radicals reacting with $HO_2$, which
correspondingly decreases the fraction that react via the OH-recycling isomerization channel, reducing OH over the Ama-
zon (-1%) and the tropospheric burdens of organic products from the isomerization pathway (e.g. $C_5$ hydroperoxyaldehydes,
-2.2%).

The cumulative effects of the changes in reactive nitrogen species are shown in Figure 2, including the absolute change in
$NO_x$ from the base to the updated mechanism (top left) and relative changes for individual species. Maps of absolute changes
of the other $NO_z$ species, which tend to highlight differences in $NO_x$-rich areas with high anthropogenic influence, are shown
in Figure S11. The net effect of the updates on the tropospheric $NO_x$ burden is a 1.7% decrease, with a stronger reduction
of NO (-2.4%) than $NO_2$ (-1.0%). This decrease is dominated by annual average reductions of over 25 ppt over China and
the Indo-Gangentic Plain as well as in the tropical upper troposphere, slightly offset by surface $NO_x$ increases of 1-10 ppt in
low-$NO_x$ regions such as the Amazon and high-latitude oceans. Both NO and $NO_2$ exhibit strong stratospheric increases (+8%
and +7% respectively), driven in the upper stratosphere by weaker $NO_x$ loss from a higher N + $O_2$ rate coefficient (N originates





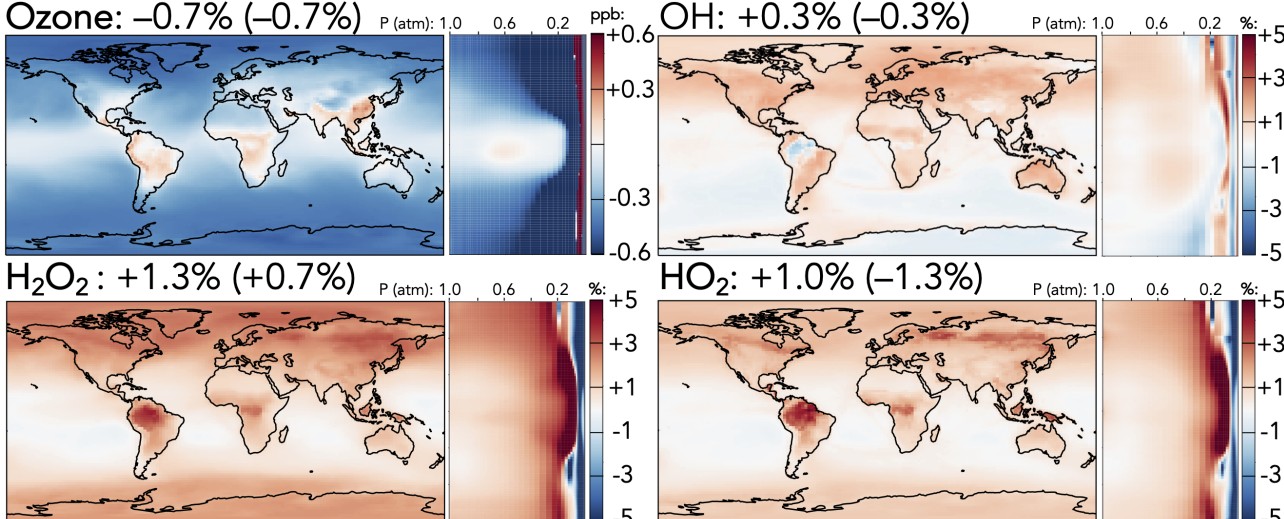

**Figure 1.** Absolute (top left, for ozone) and relative (all others) changes in the annual average mixing ratios of $(H)O_y$ species between the base and updated mechanism. Maps show surface values; atmospheric cross-sections show zonal means using the labelled altitude scale and the same latitude scales as the maps to their left. Additional vertical profiles of these species' changes can be found in Figure S12 in the supplementary material. Scales differ between species but are the same for each individual species' surface maps and cross-sections. Numbers next to species names show the percent change in their annual average tropospheric burden (stratospheric burden in parentheses) from the base to the updated mechanism.

from NO photolysis and the $N + O_2$ reaction returning NO competes with $N + NO$ producing $N_2$) and in the lower stratosphere by changes to the $HNO_z + OH$ reactions.

Other $NO_z$ species generally exhibit changes directly attributable to updates in their own sources or sinks. $NO_3$, for which loss rate coefficients to reactions with $NO_x$ and OH are revised downward and the stratospheric source from $O + NO_2$ is revised upward, generally increases at the surface (up to +0.5 ppt in India and China) and in the stratosphere (+0.4%), although this is offset by a decrease in the mid-troposphere. Changes in $N_2O_5$ follow those of its precursors, $NO_2$ and $NO_3$, including a decrease of 1.5% in the troposphere and an increase of 6.8% in the stratosphere. Nitrous acid ($HNO_2$) is influenced both by changes to its precursors ($NO_x$) and a decrease to its loss rate coefficient via reaction with OH; as a result, it decreases in the troposphere (-1.3%), especially over India and China where decreases to $NO_x$ are strongest, and increases in the stratosphere (+2.5%). Nitric ($HNO_3$) and pernitric ($HNO_4$) acids increase slightly in both the troposphere and stratosphere, due to the combined effects of changes in their OH loss rate coefficients and in the mixing ratios of their major precursors.

Among $C_1$ species, whose changes are shown in Figure 3, two effects dominate: the increased $CO + OH$ rate coefficient and the decreased $C_1$ stabilized Criegee Intermediate ($CH_2OO$) + $H_2O$ rate coefficient. The former causes a 2.7% decrease in the tropospheric CO burden, corresponding to 1-2 ppb in the Southern Hemisphere and 2-3 ppb in the Northern Hemisphere, with smaller decreases at the continental surface where the change is partially offset by increased CO production from faster







**Figure 2.** Absolute (top left, for $NO_x$) and relative (all others) changes in the annual average mixing ratios of $NO_z$ species between the base and updated mechanism. Maps show surface values; atmospheric cross-sections show zonal means using the labelled altitude scale and the same latitude scales as the maps to their left. Additional vertical profiles of these species' changes can be found in Figure S13 in the supplementary material. Scales differ between species but are the same for each individual species' surface maps and cross-sections. Numbers next to species names show the percent change in their annual average tropospheric burden (stratospheric burden in parentheses) from the base to the updated mechanism.

oxidation of volatile organics. The latter sharply reduces the formation of hydroxymethyl hydroperoxide (HMHP), the major product of $CH_2OO + H_2O$, decreasing its tropospheric burden by 22%, with the strongest absolute effects (-50-75 ppt) over tropical and mid-latitude forests where the ozonolysis of biogenic emissions leads to high $CH_2OO$ production. Formic acid, a product of the competing reaction of $CH_2OO$ with the water dimer, increases accordingly by 3.1%. The tropospheric burdens





of other $C_1$ species increase by 0.1-1.3% due to indirect effects of faster organic oxidation from the mechanism updates, with the strongest effects at the surface in regions of high biogenic influence and in the tropical upper troposphere.



**Figure 3.** Relative changes in the annual average mixing ratios of $C_1$ species between the base and updated mechanism. Maps show surface values; atmospheric cross-sections show zonal means using the labelled altitude scale and the same latitude scales as the maps to their left. Additional vertical profiles of these species' changes can be found in Figure S14 in the supplementary material. Scales differ between species but are the same for each individual species' surface maps and cross-sections. Numbers next to species names show the percent change in their annual average tropospheric burden from the base to the updated mechanism.

Figure 4 shows the effects of the mechanism updates on selected larger organic species. Two effects dominate: the reduced PAA + OH rate coefficient and the rebalanced coefficients for primary versus secondary hydrogen abstract from propane by OH. The former leads directly to a 41% increase in the tropospheric PAA burden, with the strongest effects (+150%) in the





upper troposphere where the longer PAA lifetime enables greater transport from lower-tropospheric source regions. Further, reduced regeneration of the peroxyacetyl radical (the product of PAA + OH) is compounded by an increase in its self-reaction rate coefficient; as a result, the products of other peroxyacetyl radical reactions are all decreased, including PAN (-3.8%) and acetic acid (-3.4%). These reductions are enhanced by the updates to PAN cycling rate coefficients and the increased $C_2$ Criegee Intermediate + $SO_2$ coefficient, which decreases acetic acid formation via the competing $C_2$ Criegee Intermediate +

$H_2O$ reaction. The strongest relative effects of these changes (PAN -12%, acetic acid -6%) are over the tropical oceans, while their strongest absolute effects (-20 and -10 ppt respectively) are over tropical forests.

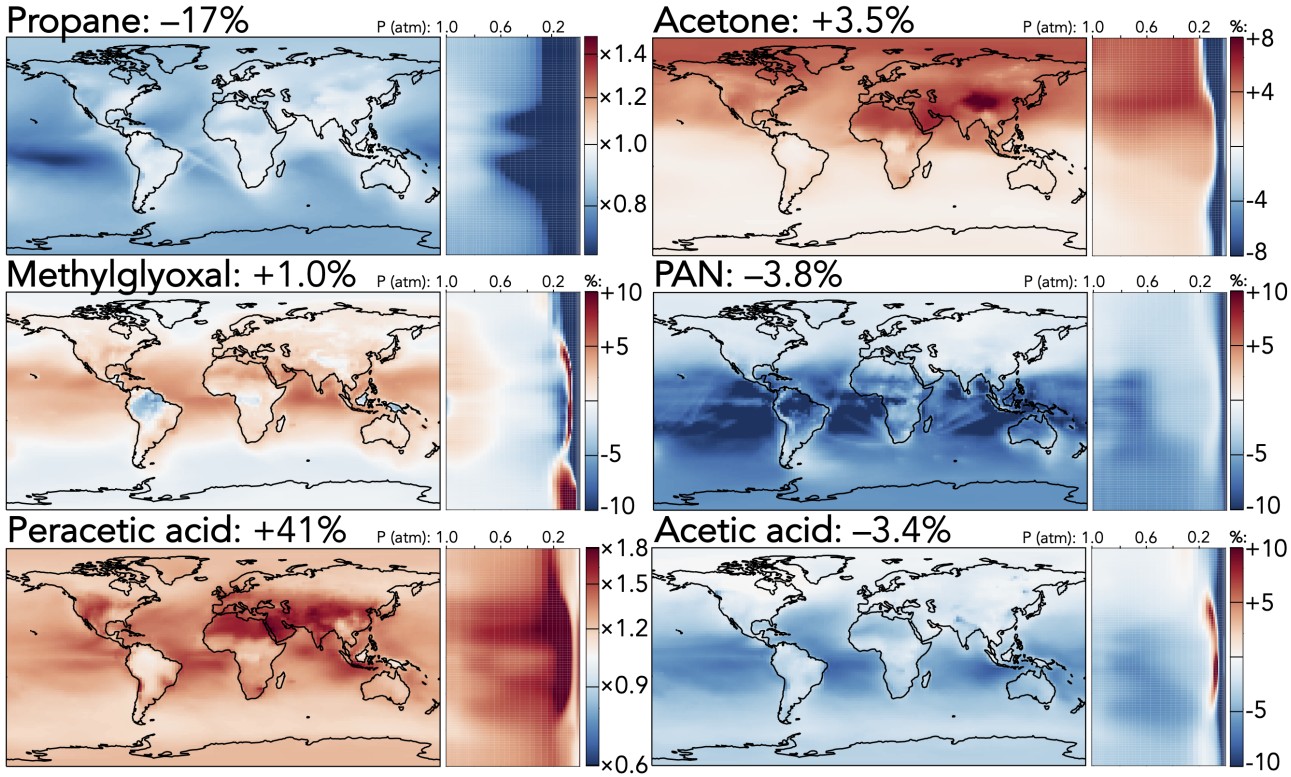

**Figure 4.** Relative changes in the annual average mixing ratios of select organic species between the base and updated mechanism. Maps show surface values; atmospheric cross-sections show zonal means using the labelled altitude scale and the same latitude scales as the maps to their left. Scales differ between species but are the same for each individual species' surface maps and cross-sections. Numbers next to species names show the percent change in their annual average tropospheric burden from the base to the updated mechanism.

The adjusted propane oxidation mechanism causes a direct 17% decrease to the tropospheric propane burden due to the overall increased $C_3H_8$ + OH rate coefficient, as well as a number of knock-on effects from the increased proportion of secondary hydrogen abstraction relative to primary. Tropospheric burdens of the products of the peroxy radical from secondary

abstraction all increase ($i$-propyl hydroperoxide +13%, $i$-propyl nitrate +8.6%, acetone +3.5%), while those from primary abstraction decrease ($n$-propyl hydroperoxide -4.3%, $n$-propyl nitrate -7.9%). Figure 4 shows the effects on acetone, the most



abundant and commonly measured of these products, but the spatial patterns of the other products are similar. While the relative changes in acetone burden (+4-8%) are evenly spread throughout the troposphere in the Northern Hemisphere, absolute changes are strongest over strong anthropogenic propane source regions (e.g. +200 ppt over East China). Products of acetone oxidation

also exhibit enhancements; for example, the decrease in PAN mixing ratios from the updated PAA + OH coefficient is offset by the increase from higher acetone production, leading to minimal change in the northern extratropics, and the tropospheric methylglyoxal burden increases by 1%. Methylglyoxal is also influence by the updated temperature dependence of its reaction with OH, leading to higher loss rates (and therefore lower mixing ratios) at colder temperatures, e.g. at high latitudes, and lower loss rates in the warmer tropics.

Finally, Figure 5 shows the effects of the mechanism updates on selected halogen radical species, which are dominated by changes to the participation of IO in the chlorine and bromine cycles. In general, the decreased IO + ClO rate coefficient leads to higher chlorine reactivity via other pathways and a general increase in burdens of chlorine radical species; the tropospheric burden of ClO rises by 4.1% and that of $Cl_2O_2$, the major product of ClO + ClO, by 27%, with the strongest effects in the upper troposphere and over the Antarctic. Other chlorine radical species exhibit smaller increases (Cl +1.4%, ClOO +3.0%)

with similar spatial patterns, although OClO, a major product of IO + ClO, decreases (-0.7%). Effects in the stratosphere are similar except that ClO and $Cl_2O_2$ decrease, likely due to higher losses from increased $NO_x$.

In contrast to the effects on chlorine, iodine and bromine radical burdens are generally slightly decreased in the troposphere due both to the updated iodine chemistry with ClO/BrO/$NO_x$ and to changes in $NO_x$ distributions. Tropospheric burdens of most iodine species are moderately lower (I -1.5%, IO -0.3%, $I_2O_4$ -5.8%, INO -6.0%, IONO -9.2%), while those of bromine

species are less affected (Br -0.6%, BrO -0.7%, $BrNO_2$ -2.9%, $BrNO_3$ -1.5%), although spatial variability due to adjusted temperature dependence and $NO_x$ patterns lead to local changes of up to 3% in Br and BrO mixing ratios. In the stratosphere, colder temperatures lead to an increase in the IO + BrO rate coefficient, causing the stratospheric BrO burden to decrease (-1.3%) and those of the IO + BrO products to increase (Br +1.0%, OIO +9.7%).

Among the changes not shown in Figures 1-5, most are minor or are attributable to changes in carbon/nitrogen conservation

(Table S2). Despite revisions to the rate coefficients of their reactions with OH, the atmospheric burdens of $SO_2$ and OCS barely change (+0.6% and -0.04% respectively). The addition of a strong temperature dependence to the isoprene ozonolysis rate coefficient also makes little difference; the tropospheric isoprene burden increases by 0.9%, and the decreased fraction of isoprene lost to ozonolysis contributes to the reduced HMHP described above. Most other volatile organic compounds exhibit slight decreases, likely due to the increase in tropospheric OH; for example, the tropospheric burdens of benzene,

toluene, xylene, and lumped $C_{4+}$ alkanes decrease by 0.5–0.6%. The effects on ethylene (-1.8%) and methacrolein (-2.1%) are enhanced by their increased ozonolysis rate coefficients. Carbon accounting changes cause larger increases to nonspecific $C_{3+}$ organic products, e.g. RCHO ($C_{3+}$ aldehydes; +7.4%), ROH ($C_{3+}$ alcohols; +21%), and RP ($C_{3+}$ hydroperoxides; +14%). Finally, the correction of an error in the MVKN yield from its precursor peroxy radical + NO reaction, which had previously caused a net loss of reactive nitrogen from the mechanism, increases the tropospheric burden of MVKN – a species typically

underestimated by models (Tsiligiannis et al., 2022) – by 39%.



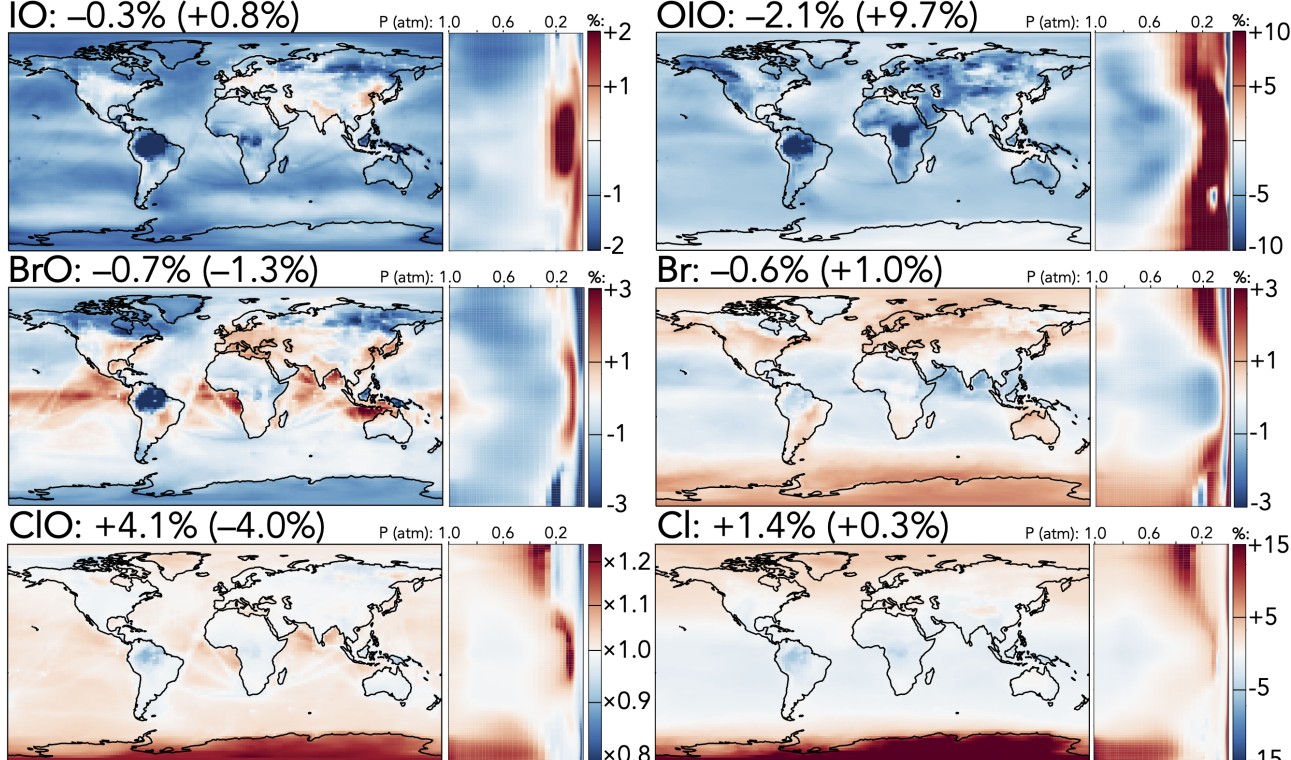

**Figure 5.** Relative changes in the annual average mixing ratios of select halogen species between the base and updated mechanism. Maps show surface values; atmospheric cross-sections show zonal means using the labelled altitude scale and the same latitude scales as the maps to their left. Additional vertical profiles of these species' changes can be found in Figure S15 in the supplementary material. Scales differ between species but are the same for each individual species' surface maps and cross-sections. Numbers next to species names show the percent change in their annual average tropospheric burden (stratospheric burden in parentheses) from the base to the updated mechanism.

## 5 Conclusions

We updated the chemical mechanism of the GEOS-Chem atmospheric chemistry model, used by a large research community for a wide range of applications, with rate coefficient and product branching ratio from recent kinetic data evaluations. A total of 63 reactions were changed, including 58 on the basis of the 2020 JPL Data Evaluation and 5 on the basis of recent IUPAC recommendations. We further updated 63 reactions to improve carbon and nitrogen conservation between reactants and products. We then quantified the effects of these updates on the GEOS-Chem simulation with reference to version 14.0.1 of the model.

Among the most notable changes to the tropospheric burdens of organic species are a 17% decrease in propane due to updates to its OH chemistry, with accompanying changes to its oxidation products (e.g. a 3.5% increase to acetone); a 41% increase in PAA due to its decreased reaction rate coefficient with OH, with accompanying decreases in its downstream products of PAN



(-3.8%) and acetic acid (-3.4%); and a 22% reduction in HMHP due to a reduction in the $CH_2OO + H_2O$ rate coefficient, along with a corresponding 3.1% increase in formic acid produced via a competing pathway. A reformulation of the rate coefficient for the CO + OH reaction leads to a 2.7% reduction in the tropospheric CO burden. Eleven updates to reactions of reactive nitrogen species result in a 1.7% net decrease in tropospheric $NO_x$ and a 7% net increase in the stratospheric $NO_x$ burden,

with smaller changes to other $NO_z$ species. Updates to the rate coefficients of iodine cycling reactions, especially those of IO with ClO, BrO, and NO, cause minor changes to halogen radical distributions, with general small increases in chlorine radical species and decreases in iodine and bromine species. Most secondary effects on trace gases of general wider interest are also minor, including slight reductions in tropospheric burdens of ozone (-0.7%) and increased loadings of formaldehyde (+0.7%) and OH (+0.3%), the latter of which increases the methane loss rate by a corresponding 0.4%. The updated mechanism

will more accurately simulate the complex chemical dependencies of key atmospheric species on temperature, pressure, and concentrations of other compounds, and the improved conservation of carbon and nitrogen will facilitate future studies of their overall atmospheric budgets.

*Code and data availability.* KPP (Lin et al. (2023), available at https://kpp.readthedocs.io/, last accessed 24 April 2023) and GEOS-Chem (https://geoschem.github.io/, last accessed 24 April 2023; DOI: 10.5281/zenodo.7271974) are both available online for public use. GEOS-

Chem mechanism inputs and simulation outputs for this work are available online at "Code and files for 'Technical note: Impacts of updated rate constants on tropospheric composition'", https://doi.org/10.7910/DVN/IDYV3E, Harvard Dataverse, V2, last accessed 30 April 2023.

*Author contributions.* KHB, MJE, and BHH conceived and planned this research; KHB performed simulations and analysis; all authors contributed to manuscript preparation.

*Competing interests.* The authors declare no competing interests.

*Acknowledgements.* KHB and DJJ acknowledge funding from the U.S. Environmental Protection Agency (EPA) Science to Achieve Results (STAR) grant program (R840014). The views and opinions expressed in this article are those of the authors and do not represent the official views of the EPA.



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
