# Peer review of "Impacts of updated reaction kinetics on the global GEOS-Chem simulation of atmospheric chemistry"

_EGUsphere, 2023_

## Author Comment (AC1)

15 December 2023

We thank the reviewers for their careful consideration of our manuscript and their helpful comments. We have reproduced the comments in their entirety below and have addressed each question and concern individually. Reviewer comments are written in black and our responses are in red; text from the manuscript is shown *italicized* with new text added to the manuscript *italicized and underlined*.

Reviewer 1:

"In this manuscript the authors present the results of their efforts to update the chemical mechanism of the GEOS-Chem chemical transport model, bringing its simulated chemistry more inline with recent recommendations from JPL and IUPAC sources. These updates represent mostly incremental progress towards mechanism improvements, including updated rate constants, temperature dependence, and branching ratios, as well as steps towards the conservation of modeled carbon and nitrogen. Impacts on annual budgets of key trace gas species appear to be mostly subtle, but certainly non-negligible, and likely have even more pronounced impacts at finer spatiotemporal scales. This effort represents an important step towards ongoing model maintenance and improvement, and will benefit the current and future modeling studies whose findings rely on accurate representation of the species included in these updates. On the whole I find this paper to be straightforward, well-written, and ready for publication. Manuscript text is clear and concise, figures are effective and well-designed, and I have no concerns regarding the authors' methods or decisions."

We appreciate the reviewer's thoughtful consideration and are delighted to hear that they find the manuscript suitable for publication as is!

Reviewer 2:

"This manuscript describes recent updates to the gas-phase thermal reaction rate coefficients in the commonly used GEOS-Chem chemical transport model to match those of the most recent NASA JPL and IUPAC recommendations. Of particular value, the authors also report improvements in the stoichiometric closure of carbon and nitrogen in the non-elementary organic oxidation reactions. While acknowledging there is still work to be done in this regard, it is a significant and important update for helping to achieve mass conservation and interpret chemical budgets in the model. The manuscript describes changes in the key gaseous chemical species in the model for the troposphere and stratosphere. It reports non-zero but generally small changes in abundance, although there are some interesting outliers (e.g., an ~20% decrease in tropospheric propane). The manuscript is well-written and clear, and I recommend publishing it, pending some minor suggestions below."

"I wish the paper had some discussion of the differences with respect to atmospheric observations. For example, do the objective improvements in the rate constants move the spatial and temporal correlation of ozone in the model away from or closer to the TOAR climatological products (doi:10.1594/PANGAEA.876108)? However, I realize it is within the scope of GMD to allow the publication of development and technical papers without evaluation."

We thank the reviewer for their careful scrutiny of our paper and their helpful suggestions detailed below. While we acknowledge that model-measurement comparisons of key species such as ozone are invaluable for model validation and benchmarking, we agree with the reviewer that such comparisons are not necessary for a GMD manuscript and consider this type of evaluation beyond the limited scope of this work. As the reviewer notes above, the updates to rate coefficients made here cause only minor changes to most species' tropospheric abundance, well within the bounds of uncertainty of most measurements. For those species which do see substantial changes, such as propane, model-measurement disparities are unlikely to be driven exclusively by reaction rate coefficients; for example, a recent analysis showed that respeciating the default GEOS-Chem global anthropogenic emission inventory based on regional inventories increased the tropospheric propane burden by 52%, though the model remains biased low relative to observations (DOI: 10.5194/egusphere-2023-2557). The rate coefficient updates implemented here represent a valuable step forward for the model regardless of their effects on model bias. For these reasons, we decided not to pursue an extensive model-measurement comparison within this study.

"I was curious when reading the manuscript if the authors ever found discrepancies between the JPL and IUPAC recommendations, and if so, how did they prioritize one over the other? It would be helpful to add a sentence to that effect."

Because GEOS-Chem has historically adhered more closely to the JPL recommendations, and used other NASA products such as GEOS meteorology, we opted to use rate coefficients from the JPL data evaluation whenever possible. Rate coefficients from IUPAC are only used for reactions that aren't evaluated in the JPL recommendations, which is why so few of them appear in this manuscript. This typically applies only to reactions of oxygenated organic compounds, such as the reactions of hydroxyacetone, methyl ethyl ketone, and methylglyoxal with OH. We have added a sentence to the manuscript to clarify the prioritization of JPL over IUPAC recommendations:

*Here we focus on gas-phase thermal chemistry updates as given by JPL and IUPAC. Rate coefficients from the latest JPL recommendations are prioritized over those from IUPAC, the latter of which are only implemented for reactions of oxygenated organics not included in the JPL data evaluation.*

"The simulations used were initialized for one year, and then one year was used for analysis in both the troposphere and stratosphere. It would be important to explicitly mention in the vicinity of Section 3 Paragraph 2 that these simulations only reflect changes from the rate constants on such time scales, and the stratospheric results should be taken through such a lens. In reality, the changes in the stratospheric (and even tropospheric) abundances of many of these species would be different than reported here if both simulations were allowed to achieve equilibrium with respect to stratospheric-tropospheric exchange, i.e., they were initialized for 10-15 years."

We thank the reviewer for pointing out this important consideration; we have added a sentence to the second paragraph of Section 3 to clarify this point:

*While this spinup is sufficient to demonstrate direct changes to most species concentrations from updated rate coefficients and product yields, it does not reflect effects of longer-term processes like stratosphere-troposphere exchange.*

"L30 - corrupted BibTeX reference"

The reference has been corrected to read *Lin et al., 2021*.

"S2 - It would be useful for readers, especially new GEOS-Chem users, if the authors could include a brief summary of the key historical papers/updates to the GEOS-Chem chemical mechanism up until this work."

A full accounting of the history of GEOS-Chem's chemical mechanism is made quite difficult by the fact that most updates are piecemeal and a few reactions at a time. Unfortunately, previous revisions to match the JPL recommendations – updates added February 2013 for JPL publication 10-6 and March 2017 for JPL publication 15-10 – were not detailed in any single manuscript, and so cannot be cited here. We have, however, expanded our description of the GEOS-Chem mechanism here to provide more detail on the timeline of updates, and how the JPL recommendations fit into that timeline. The first paragraph of Section 2 now reads as follows:

"*The GEOS-Chem model includes a detailed mechanism for oxidant-aerosol chemistry in the troposphere and stratosphere. Since its first iteration as a model of tropospheric oxidant chemistry (Bey et al., 2011), the standard GEOS-Chem chemical mechanism has used kinetics and products based on JPL recommendations (DeMore et al., 1997). The model was later expanded to include aerosol chemistry (Park et al., 2004) and stratospheric chemistry (Eastham et al., 2014), and reaction rates and products were updated based on subsequent JPL data evaluations (Sander et al., 2006, 2011; Burkholder et al., 2015).* Photolysis frequencies are calculated with the Fast-JX code (Bian and Prather, 2002) as implemented in GEOS-Chem by Mao et al. (2010) for the troposphere and Eastham et al. (2014) for the stratosphere. *Subsequent additions to the mechanism, such as detailed halogen chemistry (initially from Sherwen et al., 2016 and Chen et al., 2017, subsequently updated by Wang et al., 2021) as well as schemes for the oxidation of isoprene (initially from Mao et al., 2013, updated by Marais et al., 2016 and Bates and Jacob, 2019), monoterpenes (Fisher et al., 2016), and aromatics (Bates et al., 2021), include reactions not assessed in the JPL data evaluations; kinetics and products of these reactions use recommendations from IUPAC (Atkinson et al., 2006) or other sources detailed in the publications describing each update. More information on the GEOS-Chem mechanism can be found in the model documentation at http://www.geos-chem.org.*"

"L51 - Please define the $NO_z$ family"

We meant this to say $NO_y$, so we thank the reviewer for pointing this out! We have changed "*reactive nitrogen ($NO_z$)*" to say "*reactive nitrogen ($NO_y = NO + NO_2$ + all oxidized odd nitrogen species)*". We have also added a definition of $NO_x$ ("*$NO + NO_2$*") to the first instance where it appears in the manuscript's main text.

"L56 - Erroneous extra of"

The extraneous "of" has been removed.

"L57 - Typo in resulting an increases"

The erroneous "an" has been changed to *in*.

"L143 - van der Werf et al. 2017 is a more direct reference for GFED4"

The citation has been changed to the 2017 paper.

"S3 Model Description - should probably include the natural $NO_x$ sources and marine organic sources for completeness"

The list of emissions included in Section 3 was not meant to be exhaustive, but rather to outline the three major categories (anthropogenic, biogenic, and biomass burning) in the default GEOS-Chem emission settings. A complete listing would have to include a great deal more detail – not just the natural $NO_x$ and marine organic sources that the reviewer correctly points out, but separate aircraft and ship emissions, volcanic emissions, dust aerosols, sea salt aerosols, trash burning, biofuels, and others. This level of detail is not usually included in manuscripts' descriptions of their model setup, and is rather available online, and accessible to readers with the model version DOI provided earlier in the paragraph. To clarify these points, we have added the following sentence to the first paragraph of Section 3:

*Additional emission sources, including air-sea exchange of organics, NOx from soils and lightning, and dust and sea salt aerosol, follow the default GEOS-Chem settings.*

"L160 - HNOz should have a subscript"

The subscript has been added.

"L183 - typo in Indo-Gangentic"

The typo has been corrected to read *Indo-Gangetic*.

Reviewer 3:

"The manuscript by Bates et al. describes a series of updates to reaction rate coefficients and products in the GEOS-Chem model to reflect the most recent recommendations in a pair of widely used evaluations of chemical kinetics, the JPL and IUPAC reports. Additional modifications to improve the conservation of carbon and nitrogen in the chemical reactions are also described. The effects of the modifications on the distributions of a number of key chemical species in GEOS-Chem are then analysed using output from a one-year run. The GEOS-Chem model is very, very widely used in the community and the description and analysis of the modifications to the gas-phase chemical scheme will be received with interest. The manuscript is very well written and the changes are clearly described, including a very extensive set of tables and graphs showing the temperature and pressure dependence of the original and updated

reaction rates in the supplementary information. I have no significant concerns with the manuscript in its present form. Although I do have a few questions about how changes in the model distribution of a couple of species are attributed that I would like to see clarified or expanded on - these are listed with other minor comments below. These questions are specific to a couple of species and do not detract from the overall strong presentation of the updates and the analysis of the effects."

We appreciate the reviewer's detailed consideration of the manuscript, and hope that the point-by-point additions and clarifications provided below are helpful.

"Lines 29 - 30: There is a question mark in the list of references (Hu et al., 2018; ?; Fritz et al., 2022)."

The reference has been corrected to read *Lin et al., 2021*.

"Line 38: The wording of 'The mechanism goes beyond these recommendations.' is a bit open to interpretation. I understand the authors mean to say the reactions included in the mechanism for isoprene, monoterpenes, etc. are not all covered in the JPL and IUPAC recommendations and, if this is what is meant, it could be more clearly stated."

Based on the recommendations of Reviewer 2 (see above), we have updated this whole paragraph, which hopefully resolves the confusion. It now reads as follows:
"*The GEOS-Chem model includes a detailed mechanism for oxidant-aerosol chemistry in the troposphere and stratosphere. Since its first iteration as a model of tropospheric oxidant chemistry (Bey et al., 2011), the standard GEOS-Chem chemical mechanism has used kinetics and products based on JPL recommendations (DeMore et al., 1997). The model was later expanded to include aerosol chemistry (Park et al., 2004) and stratospheric chemistry (Eastham et al., 2014), and reaction rates and products were updated based on subsequent JPL data evaluations (Sander et al., 2006, 2011; Burkholder et al., 2015). Photolysis frequencies are calculated with the Fast-JX code (Bian and Prather, 2002) as implemented in GEOS-Chem by Mao et al. (2010) for the troposphere and Eastham et al. (2014) for the stratosphere. Subsequent additions to the mechanism, such as detailed halogen chemistry (initially from Sherwen et al., 2016 and Chen et al., 2017, subsequently updated by Wang et al., 2021) as well as schemes for the oxidation of isoprene (initially from Mao et al., 2013, updated by Marais et al., 2016 and Bates and Jacob, 2019), monoterpenes (Fisher et al., 2016), and aromatics (Bates et al., 2021), include reactions not assessed in the JPL data evaluations; kinetics and products of these reactions use recommendations from IUPAC (Atkinson et al., 2006) or other sources detailed in the publications describing each update. More information on the GEOS-Chem mechanism can be found in the model documentation at http://www.geos-chem.org.*"

"Line 65: The change to the rate of NO2 + NO3 is described as resulting in a strong temperature dependence ['stronger temperature dependencies for NO2 + NO3 (-33% - -25%)'] but the range is given as -33% to -25%, so it seems like the more significant change is a reduction in the rate - at least at atmospheric relevant temperatures?"

We agree that the rate coefficient reduction is the more important piece of this change, and to highlight this aspect, we have therefore moved mention of this update earlier in the paragraph to

go along with the other increases and decreases: "*Other updates in this family include a rate coefficient increase for $NO_3 + O$ (+30%) and decreases for $OH + NO_3$ (-9%) and $NO_2 + NO_3$ (-33% – -25% over the range 200–300 K)*"

"Line 111: The reaction of the CH2OO criegee intermediate with water vapour is mentioned as being updated, having been included in the latest JPL recommendations. Does GEOS-CHEM account for the reaction with the H2O dimer, which seems to account for a significant amount of the reaction with water vapour? I will note that the reaction rate constant given in Table 1 is for the reaction with the monomer."

Yes, the standard GEOS-Chem mechanism does include the reaction of the $C_1$ Criegee Intermediate with the water dimer. This was added as part of the isoprene updates in model version 12.8 (DOI 10.5281/zenodo.3784796) based on Bates & Jacob 2019 (DOI: 10.5194/acp-19-9613-2019), and does indeed account for a substantial fraction of the overall $CH_2OO$ fate, as mentioned in the manuscript when discussing the outcomes of the revised $CH_2OO + H_2O$ monomer rate coefficient: "*Formic acid, a product of the competing reaction of $CH_2OO$ with the water dimer, increases accordingly by 3.1%.*"

"Lines 149 - 150: Because of the extension into the stratosphere and the long timescales for transport in that region, are the initial conditions provided at January 1, 2017 'spun up' to have a fairly realistic distribution through the stratosphere?"

While it is true that this year-long spinup would be insufficient to initialize and transport some species effectively throughout the stratosphere, the fields of species concentrations used to initialize the model on January 1, 2017 are provided with a generic GEOS-Chem "restart file" that does indeed reflect realistic distributions of model species throughout the troposphere and stratosphere. We have clarified this point with a minor edit to Section 3: "*Both simulations are performed at 2˚ x 2.5˚ horizontal resolution over 72 vertical levels, are initialized from the same generic concentration fields that reflect realistic atmospheric concentrations of model species, and are run for two years (1 January 2017 to 1 January 2019).*"

"Lines 164 - 165: Could the authors expand on the explanation for the decreases in upper tropospheric ozone that are attributed to 'higher NOx-driven losses brought about by changes to the HNOz + OH (z=2-4) reactions'? The heart of the problem is that I am not sure what exactly is meant by 'NOx-driven losses' and it is not readily apparent to me how, for example, the much faster rate for OH + HNO2 -> NO2 + H2O should result in decreased ozone through 'NOx-driven losses'. There is a decrease in NOx in the upper troposphere shown in Figure 2a, but I would have thought ozone production would be NOx limited in this region so the decreased ozone would be due to a decrease in ozone production?"

Our brief explanation here was an attempt to lump together multiple causes for changing ozone in different layers of the atmosphere, which we now see was confusing. The reviewer is indeed correct that the upper tropospheric changes correspond to areas of reduced $NO_x$ and thus reduced ozone formation, although the stratospheric ozone decrease due to catalytic losses from $NO_x$ are stronger (see Figures S12-S13). We have moved mention of the $HNO_z$ changes later into the discussion of reactive nitrogen species and Figure 2, and instead focused here on the changes to $NO_x$ that represent the immediate cause of the reduced ozone: "*The upper troposphere and mid*

*stratosphere both experience much stronger ozone decreases of up to 5% in the extratropics, due respectively to lower $NO_x$-driven formation and higher $NO_x$-driven catalytic losses. These reduction are partially balanced by an ozone increase at 50-100 hPa driven by the updated N + $O_2$ reaction, leading to a net decrease of only 0.7% in stratospheric ozone.*"

"Lines 184 - 187: These are significant changes in stratospheric NOx and I fully agree with your explanation. Very interesting."

We agree with the reviewer that this was among the most interesting (and significant) outcomes of the changes to species distributions from the rate coefficient updates!

"Lines 188 - 190: Can the 0.4% increase in NO3 in the stratosphere be separated from the general 7 - 8% increase in NOx? In fact, given the increase in NOx and the increase in the rate of O + NO2 -> NO3, I would have thought the change in NO3 would have been more similar in magnitude, or larger, than the change in NOx?"

We also found this minor change in stratospheric $NO_3$ surprising given the stronger increases to stratospheric $NO_x$, but our sensitivity simulations were inconclusive in pinpointing the precise cause of this change. Most of the rate coefficient revisions would seem to lead to increased stratospheric $NO_3$ given the higher $NO_x$ due either to stronger source terms (faster O + $NO_2$, OH + $HNO_3$) or weaker sink terms (slower OH + $NO_3$, $NO_2$ + $NO_3$, NO + $NO_3$), with two exceptions: the strongly increased rate coefficient for O + $NO_3$ resulting in a faster $NO_3$ sink, and the reduction in mid-stratospheric ozone (see Figure S12), resulting in a slower $O_3$ + $NO_2$ source. While we cannot conclusively diagnose the relative strength of these two changes, we have added a sentence to the manuscript to point them out: "*The stratospheric $NO_3$ increase is not nearly as strong as that of $NO_x$, potentially due to faster loss from the increased O + $NO_3$ rate coefficient and decreased formation via $O_3$ + $NO_2$ from the reduction in stratospheric ozone.*"